# Ferlin Overview: From Membrane to Cancer Biology

**DOI:** 10.3390/cells8090954

**Published:** 2019-08-22

**Authors:** Olivier Peulen, Gilles Rademaker, Sandy Anania, Andrei Turtoi, Akeila Bellahcène, Vincent Castronovo

**Affiliations:** 1Metastasis Research Laboratory, Giga Cancer, University of Liège, B4000 Liège, Belgium; 2Tumor Microenvironment Laboratory, Institut de Recherche en Cancérologie de Montpellier, INSERM U1194, 34000 Montpellier, France; 3Institut du Cancer de Montpeiller, 34000 Montpellier, France; 4Université de Montpellier, 34000 Montpellier, France

**Keywords:** ferlin, myoferlin, dysferlin, otoferlin, C2 domain, plasma membrane

## Abstract

In mammal myocytes, endothelial cells and inner ear cells, ferlins are proteins involved in membrane processes such as fusion, recycling, endo- and exocytosis. They harbour several C2 domains allowing their interaction with phospholipids. The expression of several Ferlin genes was described as altered in several tumoural tissues. Intriguingly, beyond a simple alteration, myoferlin, otoferlin and Fer1L4 expressions were negatively correlated with patient survival in some cancer types. Therefore, it can be assumed that membrane biology is of extreme importance for cell survival and signalling, making Ferlin proteins core machinery indispensable for cancer cell adaptation to hostile environments. The evidences suggest that myoferlin, when overexpressed, enhances cancer cell proliferation, migration and metabolism by affecting various aspects of membrane biology. Targeting myoferlin using pharmacological compounds, gene transfer technology, or interfering RNA is now considered as an emerging therapeutic strategy.

## 1. Introduction

Ferlin is a family of proteins involved in vesicle fusions. To date, more than 760 articles in Pubmed refer to one of its members. Most of these publications are related to muscle biology, while less than 50 are directly related to cancer. However, the emerging idea of targeting plasma membranes [1] and the discovery of a significant correlation between Ferlin gene expression and cancer patient survival, brings attention to cancer. This review focused attention on the roles of these proteins, first in a healthy context, then in cancer.

During the maturation of spermatids to motile spermatozoa in *Caenorhabditis elegans* worm, large vesicles called membranous organelles fuse with the spermatid plasma membrane. This step requires a functional FER-1 protein encoded by the fer-1 gene (*fertilization defective-1*) [2]. When FER-1 was identified and sequenced, no other known proteins had strong resemblance to it. Subsequently, homologs were found by sequence similarity in mammals, forming a family of similar proteins now called ferlins. In humans, a first *C. elegans* fer-1 homolog gene was discovered and the protein encoded by this gene was named dysferlin [3]. Shortly after, a second human FER-1-Like gene was identified. The product of the gene was named otoferlin [4]. The human EST database mining revealed a dysferlin paralog called myoferlin [5,6]. Three new members joined the ferlin gene family: FER1L4, a pseudogene; FER1L5; and FER1L6. The main features of ferlins are summarized in Table 1.

The dysferlin mutations were involved in Limb-Girdle muscular dystrophy 2B (LGMD2B), a autosomal recessive degenerative myopathy, and in Miyoshi muscular dystrophy 1 (MMD1), a late-onset muscular dystrophy [3,7]. The otoferlin mutations were described in the non-syndromic prelingual deafness (DFNB9) and in the auditory neuropathy autosomal recessive 1 (AUNB1) [4,8,9]. Nowadays, myoferlin and the 3 last members of the ferlin family are still not linked to human genetic diseases. However, myoferlin was proposed as a modifier protein for muscular dystrophy phenotype [5] and studies of myoferlin-null mice demonstrated impaired myoblast fusion and myofiber formation during muscle development and regeneration [10]. More recently, a truncated variant of myoferlin was associated with Limb-Girdle type muscular dystrophy and cardiomyopathy [11]. Here under, this review discusses that ferlins, mainly myoferlin, are involved in neoplastic diseases and are potential therapeutic targets.

## 2. Genomic Organization of Ferlin Gene Family

Ferlin genomic organization has not been extensively investigated. Nonetheless, valuable information was obtained from sequencing and subsequent gene annotation (www.ensembl.org). In *C. elegans*, fer-1 gene is approximately 8.6 kb in length and composed of 21 exons [2]. In humans, dysferlin gene (DYSF) is composed of 55 exons [12], and encodes 19 splice variant transcripts. Otoferlin gene (OTOF) contains 47 exons and encodes 7 splice variants. One of them is retaining an intronic sequence from other locus and is not coding for protein. An alternate splicing results in a neuronal-specific domain for otoferlin, regulated by the inclusion of exon 47 [8]. Myoferlin gene (MYOF), is composed of 54 exons and encodes for 9 splice variants. Four of them are not translated to protein and the shortest retains an intronic sequence. Myoferlin promoter includes several consensus-binding sites, such as for Myc, MEF2, CEBP, Sp1, AP1, and NFAT. The latter is able to bind endogenous NFATc1 and NFATc3 [13]. FER1L5 encodes 7 splice variants obtained by the arrangement of 53 exons. Five transcripts are known to encode proteins when the 2 shortest are retaining intronic sequences and do not encode protein. FER1L6 gene is composed of 41 exons and encodes a unique transcript. The main features of ferlin genes are summarized in Table 2.

## 3. Ferlin’s Structure and Localization

*Caenorrhabditis elegans* FER-1 is a large protein rich in charged residues. Charged amino acids are distributed throughout the whole protein length such that no particularly acidic or basic domains are observed. The hydrophobicity plot described a 35 amino acid long hydrophobic region at the C-terminal end [2]. To the authors’ knowledge, it has never been experimentally demonstrated. Similarity studies suggest that this region might be a transmembrane domain. FER-1 sequence analysis with Pfam protein families database [14] revealed the existence of 4 C2 domains and several other domains.

Ferlins are proteins harboring multi-C2 domains. These structural domains are ~130 amino acid long independently folded modules found in several eukaryotic proteins. They were identified in classical Protein Kinase C (PKC) as the second conserved domain out of four. The typical C2 domain is composed of a beta-sandwich made of 8 beta-strands coordinating calcium ions, participating to their ability to bind phospholipids (for review [15]). However, some C2 domains have lost their capacity to bind calcium but still bind membranes [16]. A large variety of proteins containing C2 domains have been identified, and most of them are involved in membrane biology, such as vesicular transport (synaptotagmin), GTPase regulation (Ras GTPase activating protein) or lipid modification (phospholipase C) (for review [17]).

Human ferlin proteins harbour 5 to 7 C2 domains as described in the Pfam database (Figure 1A). According to this database, in humans, 342 proteins harbour C2 domains. However, the occurrence of multiple tandem C2 domains is uncommon. Only three vertebrate protein families contain more than two C2 domains: The multiple C2 domain and transmembrane region proteins (MCTP) [18], the E-Syt (extended synaptotagmins) [19], and the ferlins. The typical feature of a C2 domain is its ability to interact with two or three calcium ions. The prototype of this domain is the C2A contained in PKC that binds phospholipids in a calcium-dependent manner. Several other distinct C2 domain subtypes, e.g. those found in PI3K and in PTEN, do not have calcium binding abilities and instead specialize in protein-protein interactions [16,17]. In classical Ca^2+^-binding C2 domains, 5 aspartate residues are involved in the ion binding [20]. Clustal omega alignment of ferlin C2 domains with PKC and synaptotagmin I C2 domains revealed that the 5 Ca^2+^-binding aspartic acids were conserved or substituted by a glutamic acid in the C2E and C2F domains of all human paralogs (Figure 1B). The aspartic acid to glutamic acid substitution is considered as highly conservative and observed in some non-ferlin Ca^2+^-binding C2 domains [21]. Some ferlins showed more C2 domains with Ca^2+^-binding potential, e.g. dysferlin and myoferlin C2C and C2D, otoferlin C2D and fer1L6 C2D [22]. The phylogenic tree created by neighbour-joining of a Clustal omega alignment of C2 domain sequences shows that a C2 domain is more similar to others at a similar position in ortholog proteins than it is to the other C2 domains within the same protein [23]. A Clustal omega alignment reveals an evolutionary distribution of the ferlin proteins into two main subgroups (Figure 1C): The type 1 ferlins containing a DysF domain and the type-2 ferlins without the DysF domain [22]. This domain is present in yeast peroxisomal proteins where its established function is to regulate the peroxisome size and number [24]. In mammals, despite the fact that its solution structure was resolved [25] and that many pathogenic point mutations occur in this region [26,27], the function of this domain remains unknown.

Immunodetection of a myoferlin-haemagglutinin fusion protein in non-permeabilised COS-7 cells confirmed the presence of the C-terminal domain of the protein in the extracellular compartment [28], supporting the functionality of the putative trans-membrane region. The sublocalisation of ferlins was further studied, indicating robust membrane localisation for dysferlin, myoferlin and Fer1L6 while only low levels of otoferlin were at the plasma membrane and Fer1L5 was intracellular. Dysferlin and myoferlin were localised within the endo-lysosomal pathway accumulating in late endosomes and in recycling compartment. GFP-myoferlin fusion protein revealed that myoferlin was colocalized with lysosomal markers in NIH3T3 cells [29]. Otoferlin has been shown to move from the trans-Golgi network to the plasma membrane and inversely. Fer1L5 was cytosolic while Fer1L6 was detected in a specific sub-compartment of the trans-Golgi network compartment [30].

## 4. Ferlin’s Interactions with Phospholipids

Ferlins are regarded as intrinsic membrane proteins through their putative transmembrane region. However, they can also interact with membranes by other domains. Experimentally, myoferlin C2A was the single C2 domain able to bind to phospholipid vesicles. A significant presence of the negatively charged phosphatidylserine (PS) was required for this interaction. Myoferlin C2A binding to PS-containing vesicles did not occur with calcium concentration similar to the one observed in the basal physiological condition (0.1 µM). Indeed, the half-maximal binding was observed at 1 µM [31], suggesting that the C2A domain is involved in specific processes inside the cell requiring Ca^2+^ release from intracellular stock, like in Ca^2+^-regulated exocytosis. When cells are stimulated by various means, including depolarization and ligand binding, the cytosolic Ca^2+ ^ concentration increases to the concentration up to 1 µM or more [32], similar to the one required by myoferlin C2A domain to bind lipids. It appears that dysferlin C2A domain has the same binding properties as myoferlin C2A domain. However, its half-maximal lipid binding was higher (4.5 µM) [31]. A recent publication confirmed that myoferlin and dysferlin C2A domains exhibit different Ca^2+^ affinities. However, they describe myoferlin C2A domain with a lower Ca^2+^ affinity than the dysferlin homolog C2 domain, and a marginal binding of myoferlin C2A domain to phospholipid mixture containing PS [33]. The binding of dysferlin C2A to PS was confirmed and extended to several phosphoinositide monophosphates in a Ca^2+^-dependent fashion. Therrien et al. observed that all remaining dysferlin C2 domains were able to bind to PS but independently of Ca^2+^ [34]. The laurdan fluorescence emission experiments suggest that dysferlin and myoferlin contribute to increase the lipid order in lipid vesicles. The magnitude of this observation was calcium-enhanced and C2 domains within both N- and C-termini of ferlins influenced lipid packing. The experiments conducted with individual recombinant ferlin’s C2A-C domains demonstrated that all of them are able to increase lipid order [35].

The authors described in the first part of this review the conservation of the 5 Ca^2+^-binding aspartate residues in the C2D-F domains of otoferlin making them putative Ca^2+^-binding sites. In addition to its C2D-F domains, otoferlin is also able to bind Ca^2+^ via its C2B and C2C domains [36]. Despite the fact that C2A domain from otoferlin does not possess all five aspartate residues, its ability to bind Ca^2+^ is still under debate. Therrien and colleagues showed that otoferlin C2A domain can bind PS in a Ca^2+^-dependent fashion, suggesting an interaction with this ion [34]. This interaction was confirmed by a direct measure of otoferlin-binding to liposomes in the presence of Ca^2+^ (1 mM). Moreover, C2A-C domains seem to bind lipids also under calcium free conditions [36]. At the opposite, a spectroscopy analysis indicates that otoferlin C2A domain is unable to coordinate Ca^2+^ ion [37].

Floatation assays were unable to confirm the interaction between otoferlin C2A and lipids. This may be due to the presence of a shorter membrane-interacting loop at the top of the domain [37]. As for dysferlin and myoferlin, otoferlin increases lipid order in vesicles. However, its C2A does not participate to the phenomenon [35].

Ferlin proteins contain also a FerA domain recently described as a four-helix bundle fold with its own Ca^2+^-dependent phospholipid-binding activity [38].

## 5. Ferlin’s Main Functions in Non-Neoplastic Cells and Tissues

### 5.1. In Mammal Muscle Cells

Dysferlin and myoferlin have a specific temporal pattern of expression in an in vitro model of muscle development. Myoferlin was highly expressed in myoblasts that have elongated prior to fusion to syncytial myotubes. After fusion, myoferlin expression was decreased. The dysferlin expression increased concomitantly with the fusion and maturation of myotubes [31]. A proteomic analysis revealed the interacting partners of dysferlin during muscle differentiation [39]. It appeared that the number of partners decreases during the differentiation process, while the core-set of partners is large (115 proteins). Surprisingly, the dysferlin homolog myoferlin was consistently co-immunoprecipitated with dysferlin. The gene ontology analysis of the core-set proteins indicates that the highest ranked clusters are related to vesicle trafficking. In the C2C12 myoblast model, immunoprecipitation experiments showed that myoferlin interacts with the Eps15 Homology Domain 2 (EHD2) apparently through a NPF (asparagine-proline-phenylalanine) motif in its C2B domain [40]. EHD2 has been implicated in endocytic recycling. It was inferred that the interaction between EHD2 and myoferlin might indirectly regulate disassembly or reorganization of the cytoskeleton that accompanies myoblast fusion.

Dysferlin-null mice develop a slowly progressive muscular dystrophy with a loss of plasma membrane integrity. The presence of a stable and functional dystrophin–glycoprotein complex (DGC), involved in muscle injury-susceptibility when altered, suggests that dysferlin has a role in sarcolemma repair process. This was confirmed in dysferlin-null mice by a markedly delayed membrane resealing, even in the presence of Ca^2+^ [41]. Pharmacological experiments conducted in skeletal muscles demonstrated that dysferlin modulates smooth reticulum Ca^2+^ release and that in its absence injuries cause an increased ryanodine receptor (RyR1)-mediated Ca^2+^ leak from the smooth reticulum into the cytoplasm [42]. In the SJL/J mice model of dysferlinopathy, annexin-1 and -2 co-precipitate with muscle dysferlin and co-localise at sarcolemma in an injury-dependent manner [43]. An immunofluorescence analysis of mitochondrial respiratory chain complexes in the muscles from the patients with dysferlinopathy revealed complex I- and complex IV-deficient myofibers [44]. This report is particularly interesting in light of the dysferlin_v1 alternate transcript discovered in skeletal muscle [45] and harboring a mitochondrial importation signal [39].

Intriguingly, at the site of membrane injury, only the C-terminal extremity of dysferlin was immunodetected. It was reported than dysferlin was cleaved by calpain [46], one of its interacting proteins [39]. The cleavage generate a C-terminal fragment called mini-dysferlin_C72_ bearing two cytoplasmic C2 domains anchored by a transmembrane domain [46]. Myoferlin expression is also up regulated in damaged myofibers and in surrounding mononuclear muscle and inflammatory cells [13]. As it was observed for dysferlin, myoferlin can be cleaved by calpain to produce a mini-myoferlin module composed of the C2E and C2F domains [47].

Membrane repair requires the accumulation and fusion of vesicles with each other and with plasma membrane at the disruption point. A role for dysferlin and myoferlin in these processes is consistent with the presence of several C2 domains and with their homology with FER-1 having a role in vesicle fusion. Moreover, mini-dysferlin and mini-myoferlin bear structural resemblance to synaptotagmin, a well-known actor in synaptic vesicle fusion with the presynaptic membrane [48].

In mouse skeletal muscle, myoferlin was found at the nuclear and plasma membrane [5]. It is highly expressed in myoblasts before their fusion to myotubes [10,31] and found to be highly concentrated at the site of apposed myoblast and myotube membranes, and at site of contact between two myotubes [10]. Myoblast fusion requires a Ca^2+^ concentration increase to 1.4 µM [49], similar to the one reported for myoferlin C2A binding to phospholipids [31]. Myoferlin-null mice myoblasts show impaired fusion in vitro, producing mice with smaller muscles and smaller myofibers in vivo [10]. All together, these observations support a role for myoferlin in the maturation of myotubes and the formation of large myotubes that arise from the fusion of myoblasts to multinucleated myotubes.

Interestingly, myoferlin-null mice are unresponsive to IGF-1 for the myoblast fusion to the pre-existing myofibers. Mechanistic experiments indicate a defect in IGF-1 internalization and a redirection of the IGF1R to the lysosomal degradation pathway instead of recycling. As expected, myoferlin-null myoblasts lacked the IGF1-induced increase in AKT and MAPK activity downstream to IGFR [50].

The defects in myoblast fusion and muscle repair observed in myoferlin-null mice are reminiscent of what was reported in muscle lacking nuclear factor of activated T-cells (NFAT). Demonbreun and colleagues suggested that in injured myofibers, the membrane damages induce an intracellular increase of Ca^2+^ concentration producing a calcineurin-dependent NFAT activation and subsequent translocation to the nucleus. The activated NFAT can therefore bind to its response element on the myoferlin promoter [13].

Using HeLa and HEK293T cell lines overexpressing ADAM-12, it was discovered that myoferlin was one of the ten most abundant interacting partners of ADAM-12 [51]. Though this was discovered in an artificial overexpressing model using cancer cells, it can be considered as pertinent in the context of muscle cell repair. Indeed, ADAM-12 is a marker of skeletal muscle regeneration interacting with the actin-binding protein α-actinin-2 in the context of myoblast fusion [52].

The differentiating myoblast C2C12 expressed Fer1L5 at the protein level with an expression pattern similar to dysferlin throughout myoblast differentiation. Fer1L5 shares with myoferlin a NPF motif in its C2B domain. As in myoferlin, this motif was described as interacting with EHD2, but also with EHD1 [53].

### 5.2. In mammal Inner Ear Cells

In adult mouse cochlea, otoferlin gene expression is limited to inner hair cells (IHC) [4]. In these cells, the strongest immunostaining of otoferlin was associated with the basolateral region, where the afferent synaptic contacts are located, suggesting that otoferlin is a component of the IHC presynaptic machinery. Ultrastructural observations confirmed the association of otoferlin with the synaptic vesicles. It appears that otoferlin is not necessary for the synapse formation [54], but rather regulates the Ca^2+^-induced synaptic vesicle exocytosis [36].

At molecular level, otoferlin interacts with plasma membrane t-SNARE (*soluble N-ethylmaleimide-sensitive-factor attachment protein receptor)* proteins (syntaxin 1 and SNAP-25) in a Ca^2+^-dependent manner [54]. Supporting this discovery, both t-SNARE proteins are known to interact with synaptotagmin I, a C2 domain harbouring protein, in the context of the classical synaptic vesicles docking [55,56]. It was reported that otoferlin relies on C2F domain for its Ca^2+^-dependent interaction with t-SNARE [57,58,59]. However, others suggest a Ca^2+^-dependent interaction through the C2C, C2D, C2E and C2F domains and a Ca^2+^-independent interaction via the C2A and C2B domains. The SNARE-mediated membrane fusion was reconstituted with proteoliposomes. This assay indicates that in presence of Ca^2+^, otoferlin accelerates the fusion process [36], suggesting that otoferlin operates as a calcium-sensor for SNARE-mediated membrane fusion.

### 5.3. In Mammal Endothelial Cells

Bernatchez and colleagues reported that dysferlin and myoferlin are abundant in caveolae-enriched membrane microdomains/lipid rafts (CEM/LR) isolated from human endothelial cells and are highly expressed in mouse blood vessels [28,60]. As observed for dysferlin in muscle cells, myoferlin regulates the endothelial cell membrane resealing after physical damage. In endothelial cells, myoferlin silencing reduced or abolished the ERK-1/2, JNK or PLCγ phosphorylation by VEGF, resulting from a loss of VEGFR-2 stabilization at the membrane. Indeed, myoferlin silencing caused an increase in VEGFR2 polyubiquitination, which leads to its degradation [28]. In contrast to what was observed in myoferlin-silenced endothelial cells, dysferlin gene silencing decrease neither VEGFR2 expression nor its downstream signalling. However, dysferlin-siRNA treated endothelial cells showed a near-complete inhibition of proliferation when they were sub-confluent. The proliferation decrease seems to be due to an impaired attachment rather than to cell death, as supported by adhesion assays and PECAM-1 poly-ubiquitination that leads to its degradation. Co-immunoprecipitation and co-localisation experiments support the formation of a molecular complex between dysferlin and PECAM-1. This PECAM-1 degradation leads, in dysferlin-null mice, to a blunted VEGF-induced angiogenesis [60]. Another angiogenic tyrosine kinase receptor Tie-2 (tyrosine kinase with Ig and epidermal growth factor homology domains-2) is significantly less expressed at the plasma membrane when myoferlin is silenced in endothelial cells [61]. In this case, it appears that proteasomal degradation plays a minor role in the down regulation of the receptor. Strikingly, G-protein coupled receptors (GCPR) were unaffected by the decrease of myoferlin expression, suggesting a selective effect on receptor tyrosine kinases (RTK).

It was also reported that in endothelial cells, myoferlin is required for an efficient clathrin and caveolae/raft-dependent endocytosis, is co-localized with Dynamin-2 protein [62] and that the FASL-induced lysosome fusion to plasma membrane is mediated by dysferlin C2A domain [63].

### 5.4. Other Mammal’s Cells

Dysferlin and myoferlin are expressed in both basal and ciliated airway epithelial cells from healthy human lungs [64]. In the airway epithelial cell line (16HBE), dysferlin and myoferlin were immuno-detected at the plasma membrane, Golgi membrane and in cytoplasm but not in the nuclei. The silencing of myoferlin in these cells induces the loss of zonula occludens (ZO)-1, inducing apoptosis [64].

Myoferlin was also detected in exosomes from human eye trabecular meshwork cells [65] and in phagocytes where it participates to the fusion between lysosomes and the plasma membrane, thus promoting the release of lysosomal contents [29].

The Fer1L5 gene expression was largely restricted to the pancreas, where it was alternatively spliced by removing exon 51 [30].

## 6. Ferlins in Cancer, Potential Targets to Kill Cancer

It is clear from the data above that ferlins are consistently involved in membrane processes requiring membrane fusion, including endocytosis, exocytosis, membrane repair, recycling and remodelling. Membrane processes are of extreme importance for cell survival and signalling, making them core machinery for cancer cell adaptation to hostile environments.

Considering that ferlins have been only scarcely investigated in cancer, the authors next sought to mine publicly available databases and gain information regarding ferlin’s expression or mutation in tumors. Using the FireBrowse gene expression viewer (firebrowse.org), The Cancer Genome Atlas (TCGA) RNAseq data of all ferlin’s genes in neoplastic tissues were investigated in order to obtain a differential expression in comparison to their normal counterparts. It appears that all ferlin genes are modulated in several cancer types. Myoferlin and fer1l4 genes are more frequently up regulated than down regulated, while dysferlin, fer1l5, and fer1l6 are more frequently down regulated (Figure 2).

Experimentally, a myoferlin gene was discovered as highly expressed in several tumour tissues including the pancreas [66,67], breast [68], kidneys [68], and head and neck squamous cell carcinoma (HNSCC) [69]. This expression was confirmed at a protein level in tumour tissue and/or cell lines from the pancreas [70,71,72,73], breast [74,75], lungs [75], melanoma [75], hepatocellular carcinoma [76], HNSCC [77], clear cell renal carcinoma [78,79], and endometroid carcinoma [80]. Myoferlin was also detected at a protein level in microvesicles/exosomes derived from several cancer cells including the bladder [81], colon [82,83,84,85], ovary [86], prostate [87], breast and pancreas, where it plays a role in vesicle fusion with the recipient endothelial cells [88].

This review then explored the mutations occurring in ferlin genes in tumours using Tumorportal (http://www.tumorportal.org) [89]. Several mutations were reported in ferlin genes in a few cancer types. However, none of them were considered as significant. Survival was also analysed (Table 3) using a pan-cancer method available online (OncoLnc–http://www.oncolnc.org) and combining mRNAs, miRNAs, and lncRNAs expression [90]. Noticeably, otoferlin expression was strongly significantly correlated with survival in renal clear cell carcinoma (KIRC–*p* < 10^−5^); myoferlin expression was strongly significantly correlated with survival in brain lower grade glioma (LGG–*p* < 10^−4^) and pancreatic adenocarcinoma (PAAD–*p* < 10^−4^), and Fer1l4 expression was strongly significantly correlated with survival in bladder urothelial carcinoma (BLCA–*p* < 10^−5^) and kidney renal clear cell carcinoma (KIRC–*p* < 10^−5^). The 5 more significant correlations between ferlin’s expression and the overall survival were represented as Kaplan-Meier curves with their associated log-rank p-value (Figure 3). 

Interestingly, a recent publication points out specific single nucleotide polymorphisms in dysferlin genes as significantly associated with pancreas cancer patient survival [91]. Mining the TCGA database, a high Fer1L4 expression was reported as a predictor of a poor prognosis in glioma [92,93] and as an oncogenic driver in several human cancers [94]. However, several other publications pointed it out as a predictor of good prognosis in osteosarcoma [95], gastric cancer [96], endometrial carcinoma [97].

### 6.1. Breast Cancer and Melanoma

A mathematical model was proposed to examine the role of myoferlin in cancer cell invasion. This model confirms the experimental observation of decreased invasion of the myoferlin-null breast MDA-MB-231 cell line, and predicts that the pro-invasion effect of myoferlin may be in large partly mediated by MMPs [98]. The model was further validated in vitro suggesting a mesenchymal to epithelial transition (MET) when myoferlin was knockdown [99,100]. Using the same cell model, Blackstone and colleagues showed that myoferlin depletion increased cell adhesion to PET substrate by enhancing focal adhesion kinase (FAK) and its associated protein paxillin (PAX) phosphorylation [101]. Interestingly, myoferlin was reported as regulating the cell migration through a TGF-β1 autocrine loop [102]. Recently, related results were reported in melanoma [103]. Myoferlin expression was first correlated with vasculogenic mimicry (VM) in patients, then its in vitro depletion in A375 cell line impaired VM, migration, and invasion by decreasing MMP-2 production.

Several evidences, obtained from normal endothelial cells, indicate that myoferlin is involved in RTKs recycling (see above). Our group showed that MDA-MB-231 and -468 cells depleted for myoferlin were unable to migrate and to undergo EMT upon EGF stimulation. The authors discovered that myoferlin depletion altered the EGFR fate after ligand binding, most probably by inhibiting the non-clathrin mediated endocytosis [104]. Unexpectedly, myoferlin seemed to be physically associated with lysosomal fraction in MCF-7 cells [105], supporting its involvement in the membrane receptor recycling.

The co-localisation of myoferlin with caveolin-1 [104], the main component of caveolae considered as a metabolic hub [106] prompted our group to investigate the implication of myoferlin in energy metabolism. In this context, the authors showed in triple-negative breast cancer cells that myoferlin-silencing produces an accumulation of monounsaturated fatty acids (C16:1). Its depletion further decreased oxygen consumption switching the cell metabolism toward glycolysis [107]. This was the first report of the role of myoferlin in mitochondrial function and cell metabolism. A recent report describing the link between dysferlin mutations and mitochondrial respiratory complexes in muscular dysferlinopathy emerged (see above) [44]. It is also intriguing that dysferlin_v1 alternate transcript discovered in skeletal muscle [45] harbours a mitochondrial importation signal [39].

Several breast cancer cell lines and tissues showed a calpain-independent myoferlin cleavage, regardless of cell injuries and subsequent Ca^2+^ influx [108]. The resulting cleaved myoferlin increases ERK phosphorylation in an overexpressing HEK293 system. It would be of interest to further study the link between mini-myoferlin and KRAS mutated cancers as ERK is a mid-pathway signalling protein in this context.

### 6.2. Pancreas and Colon Cancers

In pancreas adenocarcinoma (PAAD), myoferlin is overexpressed in high grade PAAD in comparison to low grade [73]. The patients with high myoferlin PAAD had a significantly worse prognosis than those with low myoferlin PAAD, with myoferlin appearing as an independent prognosis factor. The experiments undertaken with pancreatic cell lines and siRNA-mediated silencing demonstrated that myoferlin requested to maintain a high proliferation rate. The authors reported that myoferlin is a key element in VEGF exocytosis by PAAD cell lines, correlating with microvessel density in PAAD tissue [109]. Recently, it was demonstrated that myoferlin is critical to maintain mitochondrial structure and oxidative phosphorylation [110]. This discovery was extended to colon cancer where myoferlin seemed also to protect cells from p53-driven apoptosis [111]. The concept claiming that metastatic dissemination relies on oxidative phosphorylation is broadly accepted [112,113]. Based on these reports, the authors discovered that myoferlin was overexpressed in PAAD cells with a high metastatic potential, where it controls mitochondrial respiration [114].

Recently, FER1L4 methylated DNA marker in pancreatic juice has been strongly associated with pancreatic ductal adenocarcinoma suggesting its use as a biomarker for early detection [115].

### 6.3. Lung Cancer

In mice bearing solid LLC lung tumours, the intratumoral injection of myoferlin siRNA mixed with a lipidic vector reduced the tumour volume by 73%. The observed reduction was neither the consequence of a difference in blood vessel density nor of VEGF secretion. However, a significant reduction of the proportion of the Ki67-positive cells indicated a decrease in cell proliferation [75]. Myoferlin was reported as expressed in human non-small cell lung cancer tissues where it was correlated with VEGFR2, thyroid transcription factor (TTF)-1 and transformation-related protein (p63), especially in the low stage tumours [116].

Recently, it was suggested that long non-coding RNA Fer1L4 negatively controlled proliferation and migration of lung cancer cells, probably through the PI3K/AKT pathway [117]. The same observation was made in osteosarcoma cells [118], esophageal squamous cell carcinoma [119], and hepatocellular carcinoma [120].

### 6.4. Liver Cancer

In the hepatocellular carcinoma (HCC) cell line, the silencing of the transcriptional coactivator of the serum response factor (SRF), Megakaryoblastic Leukemia 1/2 (MKL1/2), induced a reduction of myoferlin gene expression. It was shown by chromatin immunoprecipitation that MKL1/2 binds effectively to the myoferlin promoter [76]. As in other cancer types, HCC required myoferlin to proliferate and perform invasion or anchorage-independent cell growth. Its depletion enhanced EGFR phosphorylation, in agreement with the concept of myoferlin being a regulator of RTK recycling.

### 6.5. Head and Neck Cancer

A myoferlin expression pattern was investigated in oropharyngeal squamous carcinoma (OPSCC). It was reported that myoferlin was overexpressed in 50% of the cases and significantly associated with worse survival. Moreover, HPV-negative patients had significantly higher expressions of myoferlin. A subgroup survival analysis indicates the interaction between these two parameters as HPV-negative has the worst prognosis when myoferlin is highly expressed. Nuclear myoferlin expression appeared to be highly predictive of the clinical outcome and associated with IL-6 and nanog overexpression [77]. Upon HNSCC cell line stimulation with IL-6, myoferlin dissociates from EHD2 and binds activated STAT3 to drive it in the nucleus. The observation was extended to breast cancer cell lines [69].

### 6.6. Gastric Cancer

Recently, a profiling study reported that FER1L4 was a long non-coding RNA (lncRNA) strongly downregulated in gastric cancer tissue [96], in plasma from gastric cancer patients [121] and in human gastric cancer cell lines [122]. In gastric cancer tissues, FER1L4 lncRNA was associated with the tumour diameter, differentiation state, tumour classification, invasion, metastasis, TNM stage and serum CA72-4. Interestingly, the abundance of this lncRNA decreases in plasma shortly after surgery [121]. The same team reported that the FER1L4 lncRNA is a target of miR-106a-5p [122,123]. The cell depletion in FER1L4 lncRNA resulted in an increase in miR-106a-5p and in a decrease of its endogenous target PTEN, suggesting a competing endogenous RNA (ceRNA) [124] role for FER1L4 lncRNA [122]. The control of miR-106a-5p by FER1L4 lncRNA was extended to colon cancer [125] and HCC [126], while it was described over miR-18a-5p in osteosarcoma [127].

### 6.7. Gynecological Cancers

Lnc Fer1L4 was briefly investigated in ovarian cancer where it was described as downregulated in cancer cells in comparison to normal ovarian epithelial cells [128]. Interestingly, the Fer1L4 expression correlates negatively with the paclitaxel resistance and its re-expression restore the paclitaxel sensitivity through the inhibition of a MAPK signalling pathway.

## 7. Conclusions

This review clearly shows that all ferlin proteins are membrane-based molecular actors sharing structural similarities. Far beyond their well-described involvement in physiological membrane fusion, several correlations apparently link ferlins, and most particularly myoferlin, to cancer prognosis. However, further investigations are still needed to discover the direct link between myoferlin and cancer biology. Encouragingly, there are many indications that myoferlin depletion interferes with growth factor exocytosis, surface receptor fate determination, exosome composition, and metabolism, indicating the future research axes.

Self-sufficiency in growth factor signalling is a hallmark of cancer cells. Cancer cells overproduce the growth factor to stimulate unregulated proliferation in an autocrine, juxtacrine or paracrine fashions. In this context, myoferlin could be considered as a cancer growth promoter as it helps the exocytosis of the growth factors, at least VEGF. In normal cells, myoferlin was described as involved in receptor tyrosine kinase (EGFR and VEGFR) recycling or expression, allowing as such, the cell response to the growth factors. Knowing that some cancer cells exhibit mutations in tyrosine kinase receptors, which lead to a constitutive receptor activation triggering the downstream pathways, it can be speculated that myoferlin depletion could impede cell proliferation in these cases. This role was indeed described in breast cancer cells [104].

Exosomes are small extracellular vesicles released on exocytosis of multivesicular bodies filled with intraluminal vesicles. They represent an important role in intercellular communication, serving as carrier for the transfer of miRNA and proteins between cells. The exosomes are increasingly described as cancer biomarkers [129] and involved in the preparation of the tumour microenvironment [130]. Interestingly, myoferlin was demonstrated to be present in exosomes isolated from several cancer cell types. However, the biological significance of this localization has still to be investigated.

Metabolism recently integrated the hallmarks of cancer [131], and mitochondria were recognised as key players in cancer metabolism [132]. The indications that myoferlin is necessary for optimal mitochondrial function is a promising avenue in the search for an innovative therapy.

Myoferlin, being overexpressed in several cancer types, offers very promising advantages for cancer diagnosis and targeting. Targeting myoferlin at the expression or functional levels remains, however, the next challenge. Interestingly, recent studies identified new small compounds interacting with the myoferlin C2D domain and demonstrating promising anti-tumoral/metastasis properties in breast and pancreas cancer [133,134].

Gene transfer strategies have undergone profound development in recent years and this is particularly applicable for recessive disorders. The adeno-associated virus (AAV) is a non-pathogenic vector used in a treatment strategy aiming at delivering full-length dysferlin or shorter variants to skeletal muscle in dysferlin-null mice. Several well documented reports demonstrate an improvement in the outcome measures after dysferlin gene therapy [135,136,137,138]. Similar AAV vectors were used as a gene delivery system in cancer [139,140], allowing the dream of myoferlin negative-dominant delivery to cancer cells. Moreover, the sleeping beauty transposon system [141] may overcome some of the limitations associated with viral gene transfer vectors and transient non-viral gene delivery approaches that are being used in the majority of ongoing clinical trials.

## 8. Statistical Methods

The multivariate Cox regressions (Table 3) were performed with the coxph function from the R survival library. For each cancer and data type, OncoLnc attempted to construct a model with gene expression, sex, age, and grade or histology as multivariates [90]. The clinical information was obtained from TCGA and only patients who contained all the necessary clinical information were included in the analysis. The patients were split into low and high expressing according to the median gene expression.

## Figures and Tables

**Figure 1 cells-08-00954-f001:**
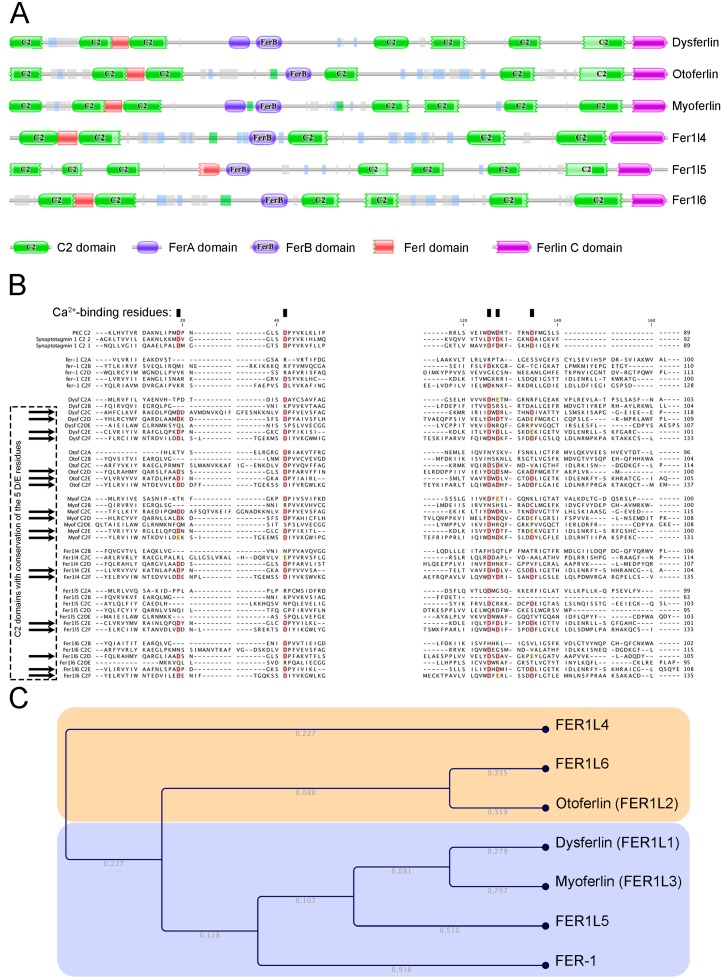
Structure and phylogenic relation of ferlin proteins. (**A**) Schematic structure of FER-1 human homologs as produced by Pfam protein families’ database. (**B**) Clustal omega multiple alignment of ferlin C2 domains. Conserved Ca^2+^-binding site are highlighted in red (aspartic acid—D) or yellow (glutamic acid—E). (**C**) Cladogram of clustal omega alignment indicating type 1 ferlins in blue and type 2 ferlins in yellow. The branch length is indicated in grey.

**Figure 2 cells-08-00954-f002:**
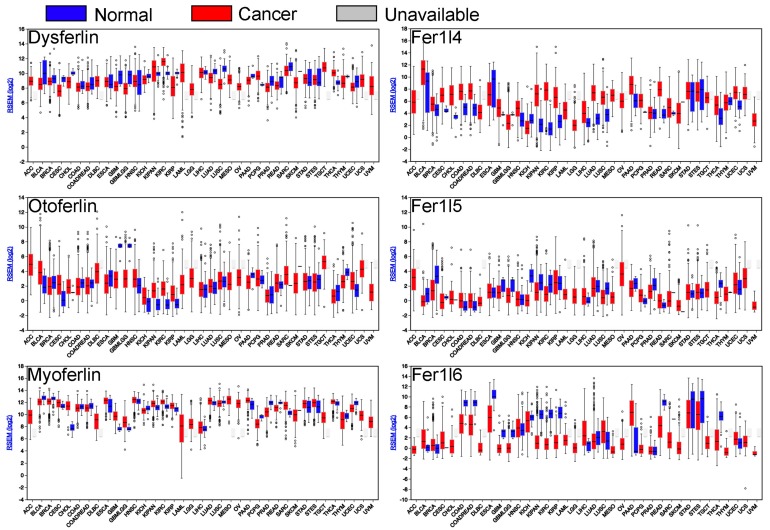
Ferlin gene expression in several cancers (red) and their normal counterparts (blue). Cancer tissues from adrenocortical carcinoma (ACC), bladder urothelial carcinoma (BLCA), breast invasive carcinoma (BRCA), cervical squamous cell carcinoma and endocervical adenocarcinoma (CESC), cholangiocarcinoma (CHOL), colon adenocarcinoma (COAD), colorectal adenocarcinoma (COADREAD), lymphoid neoplasm diffuse large B-cell lymphoma (DLBC), esophageal carcinoma (ESCA), glioblastoma multiforme (GBM), glioma (GBMLGG), head and neck squamous cell carcinoma (HNSC), kidney chromophobe (KICH), pan-kidney cohort (KIPAN), kidney renal clear cell carcinoma (KIRC), kidney renal papillary cell carcinoma (KIRP), acute myeloid leukemia (LAML), brain lower grade glioma (LGG), liver hepatocellular carcinoma (LIHC), lung adenocarcinoma (LUAD), lung squamous cell carcinoma (LUBC), mesothelioma (MESO), ovarian serous cystadenocarcinoma (OV), pancreatic adenocarcinoma (PAAD), pheochromocytoma and paraganglioma (PCPG), prostate adenocarcinoma (PRAD), rectum adenocarcinoma (READ), sarcoma (SARC), skin cutaneous melanoma (SKCM), stomach adenocarcinoma (STAD), stomach and esophageal carcinoma (STES), testicular germ cell tumours (TGCT), thyroid carcinoma (THCA), thymoma (THYM), uterine corpus endometrial carcinoma (UCEC), uterine carcinosarcoma (UCS), uveal melanoma (UVM).

**Figure 3 cells-08-00954-f003:**
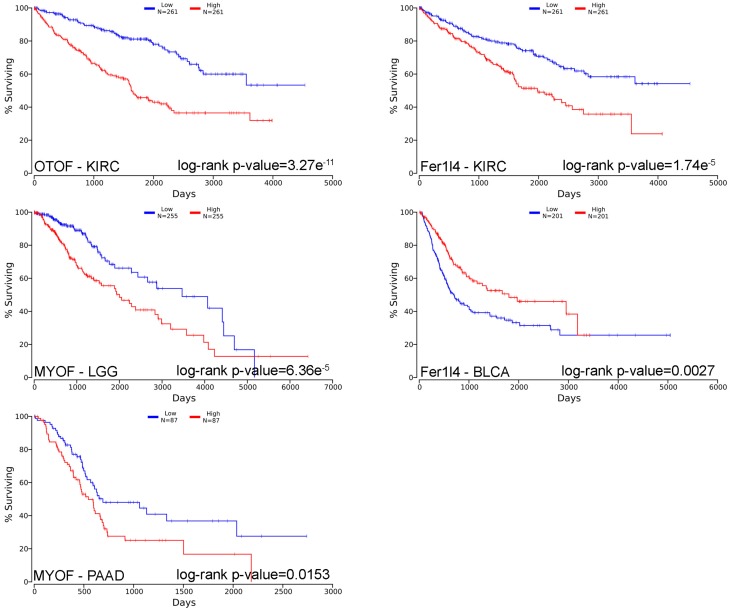
Kaplan-Meier survival curves of patient cohorts with different cancer types. Ferlin gene expression was segregated in low (blue) and high (red) expression according to median in kidney renal clear cell carcinoma (KIRC), brain lower grade glioma (LGG), bladder urothelial carcinoma (BLCA), and pancreatic adenocarcinoma (PAAD).

**Table 1 cells-08-00954-t001:** Short description of *C. elegans* and human ferlin genes and proteins.

Protein Name (Uniprot Number)	Gene Name	Chromosome Mapping	Main Protein Size
Sperm vesicle fusion protein FER-1 (Q17388)	fer-1		2034 AA (235 KDa)
Dysferlin (O75923)	Fer1-Like 1 Fer1L1	2p13.2	2080 AA (237 KDa)
Otoferlin (Q9HC10)	Fer1-Like Fer1L2	2p23.3	1997 AA (227 KDa)
Myoferlin (Q9NZM1)	Fer1-Like 3 Fer1L3	10q23.33	2061 AA (230 KDa)
FER1L4 (A9Z1Z3)	Fer1-Like 4 Fer1L4	20q11.22	pseudogene
FER1L5 (A0AVI2)	Fer1-Like 5 Fer1L5	2q11.2	2057 AA (238 KDa)
FER1L6 (Q2WGJ9)	Fer1-Like 6 Fer1L6	8q24.13	1857 AA (209 KDa)

**Table 2 cells-08-00954-t002:** Short description of *C. elegans* and human ferlin genes and transcripts.

Gene Name	Gene Length	Number of Exons	Transcript Size	Number of Variants
Fer-1	8.6 kb	21	6.2 kb	3
Fer1-Like 1 Fer1L1 (DYSF)	233 kb	55	0.5–6.7 kb	19
Fer1-Like 2 Fer1L2 (OTOF)	121 kb	47	0.5–7.2 kb	7
Fer1-Like 3 Fer1L3 (MYOF)	180 kb	54	0.4–6.7 kb	9
Fer1-Like 4 Fer1L4	48 kb	43	0.2–5.9 kb	13
Fer1-Like 5 Fer1L5	64 kb	53	3.5–6.5 kb	7
Fer1-Like 6 Fer1L6	278 kb	41	6 kb	1

**Table 3 cells-08-00954-t003:** Survival analysis by a Cox regression.

Positive Association			Negative Association		
**Cohort**	**Cox Coefficient**	**p-Value**	**Cohort**	**Cox Coefficient**	**p-Value**
DYSF EXPRESSION					
CESC	0.266	4.20e^−02^	SARC	−0.277	1.00e^−02^
STAD	0.171	4.80e^−02^	KIRC	−0.220	1.00e^−02^
OTOF EXPRESSION					
**KIRC**	**0.377**	**1.50e** ^−**06**^	BLCA	−0.275	4.50e^−04^
KIRP	0.413	4.90e^−03^	SKCM	−0.169	1.40e^−02^
MYOF EXPRESSION					
**LGG**	**0.441**	**1.40e** ^−**05**^	SKCM	−0.163	1.90e^−02^
**PAAD**	**0.561**	**1.70e** ^−**05**^			
LAML	0.215	4.70e^−02^			
FER1L4 EXPRESSION					
**KIRC**	**0.356**	**5.20e** ^−**06**^	**BLCA**	**−0.383**	**2.90e** ^−**06**^
KIRP	0.492	1.10e^−03^	SKCM	−0.225	1.10e^−03^
LGG	0.244	4.00e^−03^			
FER1L5 EXPRESSION					
LUAD	−0.199	1.30e^−02^			
FER1L6 EXPRESSION					
KIRC	−0.160	4.80e^−02^			
READ	−0.401	4.90e^−02^			

Ferlin gene expression from cohorts with cancer was submitted to a survival analysis with a Cox regression. The red rows indicate a negative Cox coefficient, the green rows indicate positive Cox coefficient. The bold p-values were considered as highly significant (*p* < 10^−4^). Bladder urothelial carcinoma (BLCA), cervical squamous cell carcinoma and endocervical adenocarcinoma (CESC), kidney renal clear cell carcinoma (KIRC), kidney renal papillary cell carcinoma (KIRP), acute myeloid leukemia (LAML), brain lower grade glioma (LGG), lung adenocarcinoma (LUAD), pancreatic adenocarcinoma (PAAD), rectum adenocarcinoma (READ), sarcoma (SARC), skin cutaneous melanoma (SKCM), stomach adenocarcinoma (STAD).

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
