# Peer review of "Ferlin Overview: From Membrane to Cancer Biology"

_cells, 2019, doi:10.3390/cells8090954_

Round 1
Reviewer 1 Report
This review provides a comprehensive (almost overwhelming) overview of ferlin biology and intriguing connections to cancer. More emphasis on the implications and significance to cancer and prognosis/outcomes should be placed early (in the abstract/introduction) to boost the significance prior to all the basic biology in the first sections. The data on ferlin expression in various cancers and correlations between ferlin expression levels and cancer prognosis/survival looks strong but no information on statistics is provided. This is technically a review article, but a section on this statistical analysis needs to be included, perhaps like a methods section at the end. Overall a very thorough and well-written review.
Minor comments:
There is not a sufficient introduction of "C2 domains". Please include adequate background on what these are and what they do.
The section on exosomes at the end seems too speculative. Consider deleting it and just mentioning the fact that ferlin can be exosomal in the main part of the biology or cancer section.
Author Response
1) This review provides a comprehensive (almost overwhelming) overview of ferlin biology and intriguing connections to cancer. More emphasis on the implications and significance to cancer and prognosis/outcomes should be placed early (in the abstract/introduction) to boost the significance prior to all the basic biology in the first sections.
Authors thank the reviewer for his/her suggestion regarding the earlier appearance of Ferlin significance in cancer. As suggested, the correlation between Ferlin expression and patient outcomes was placed in the first half of the abstract. A sentence was added in the first paragraph of introduction, indicating the emerging idea of perturbing plasma membrane in cancer therapies (including adequate reference), as well as the discovered correlation between Ferlin expression and patient survival.
2) The data on ferlin expression in various cancers and correlations between ferlin expression levels and cancer prognosis/survival looks strong but no information on statistics is provided. This is technically a review article, but a section on this statistical analysis needs to be included, perhaps like a methods section at the end. Overall a very thorough and well-written review.
Authors thank the reviewer for his/her sounding suggestion, however, statistics were not performed by authors themselves. Presented results were obtained from a web-based tool named OncoLnc (reference 90). To avoid confusion, the web address of this tool was added to the main text (Line 335 – page 9). Anyway, as requested, we add a statistic section at the end of the manuscript (section 8 – page 14). This section describes briefly how OncoLnc works and how the low- and high-expressing patiet groups were defined by authors.
Minor comments:
3) There is not a sufficient introduction of "C2 domains". Please include adequate background on what these are and what they do.
Authors thank the reviewer for this relevant comment. A paragraph describing briefly C2 domains and their functions was included in section 3 Ferlin’s structure and localization (Line 86 – page 3).
4) The section on exosomes at the end seems too speculative. Consider deleting it and just mentioning the fact that ferlin can be exosomal in the main part of the biology or cancer section.
Authors agree with reviewer’s comment. The corresponding part of the conclusion was toned down and the suggestion to use myoferlin as a target to prevent exosome release or fusion was removed.
Reviewer 2 Report
In the submitted manuscript, Peulen and colleagues systematically analyze the Ferlin family of proteins. In the first half of the manuscript, the authors revise the molecular details, structural, and cellular features of the main members of the Ferlin family: Dysferlin, Otoferlin, Myoferlin, Fer1L4, Fer1L5, and Fer1L6. In the second part of the manuscript, the Authors capitalize on the sequencing data of patients with different cancers and extract the expression profiles and up/down regulations of various Ferlin members in distinct cancer pathologies.
Overall, this is a very interesting review article that nicely brings together both known molecular details with the novel correlation analysis of Ferlin misregulation in patients with different cancers.
My concern is about the tone of the Conclusion section. While in silico correlation analysis suggest a potential link between Ferlin biology and neoplasia, there is certainly not enough of mechanistic knowledge to claim a direct link between Ferlins, cancer cell survival, cell proliferation and exchange of macromolecules with exosomes. This chapter should be rewritten more cautiously -- correlation does not imply causality. Once this concern is addressed, I endorse publishing this manuscript in Cells.
Author Response
In the submitted manuscript, Peulen and colleagues systematically analyze the Ferlin family of proteins. In the first half of the manuscript, the authors revise the molecular details, structural, and cellular features of the main members of the Ferlin family: Dysferlin, Otoferlin, Myoferlin, Fer1L4, Fer1L5, and Fer1L6. In the second part of the manuscript, the Authors capitalize on the sequencing data of patients with different cancers and extract the expression profiles and up/down regulations of various Ferlin members in distinct cancer pathologies.
Overall, this is a very interesting review article that nicely brings together both known molecular details with the novel correlation analysis of Ferlin misregulation in patients with different cancers.
1) My concern is about the tone of the Conclusion section. While in silico correlation analysis suggest a potential link between Ferlin biology and neoplasia, there is certainly not enough of mechanistic knowledge to claim a direct link between Ferlins, cancer cell survival, cell proliferation and exchange of macromolecules with exosomes. This chapter should be rewritten more cautiously -- correlation does not imply causality. Once this concern is addressed, I endorse publishing this manuscript in Cells.
Authors thank the reviewer for his/her sounding suggestion improving strongly our manuscript. Following the reviewer advice, we toned down the conclusion. First, we underlined the correlative aspect of the link between ferlin expression and patient survival. Then, paragraph concerning exosomes was amended, the suggestion to use myoferlin as a target to prevent exosome release or fusion was removed. Finally, a paragraph concerning the emergence of metabolism in cancer was included.